# Development of a Fast Liquid Chromatography Coupled to Mass Spectrometry Method (LC-MS/MS) to Determine Fourteen Lipophilic Shellfish Toxins Based on Fused–Core Technology: In-House Validation

**DOI:** 10.3390/md19110603

**Published:** 2021-10-24

**Authors:** Araceli E. Rossignoli, Carmen Mariño, Helena Martín, Juan Blanco

**Affiliations:** Centro de Investigacións Mariñas (CIMA), Pedras de Corón s/n, 36620 Vilanova de Arousa, Spain; maria.carmen.marino.cadarso@xunta.gal (C.M.); helena.martin.sanchez@xunta.gal (H.M.); juan.carlos.blanco.perez@xunta.gal (J.B.)

**Keywords:** emerging toxins, fast method, LC-MS/MS, lipophilic toxin, performance, screening

## Abstract

Prevalence and incidence of the marine toxins (paralytic, amnesic, and lipophilic toxins) including the so-called emerging toxins (these are, gymnodimines, pinnatoxins, or spirolides among others) have increased in recent years all over the world. Climate change, which is affecting the distribution of their producing phytoplankton species, is probably one of the main causes. Early detection of the toxins present in a particular area, and linking the toxins to their causative phytoplankton species are key tools to minimize the risk they pose for human consumers. The development of both types of studies requires fast and highly sensitive analytical methods. In the present work, we have developed a highly sensitive liquid chromatography-mass spectrometry methodology (LC-MS/MS), using a column with fused-core particle technology, for the determination of fourteen lipophilic toxins in a single run of 3.6 min. The performance of the method was evaluated for specificity, linearity, precision (repeatability and reproducibility) and accuracy by analysing spiked and naturally contaminated samples. The in-house validation was successful, and the limit of detection (LOD) and quantification (LOQ) for all the toxins were far below their regulatory action limits. The method is suitable to be considered in monitoring systems of bivalves for food control.

## 1. Introduction

Lipophilic toxins are natural metabolites produced by dinoflagellates which can be extracted from bivalve tissues using organic solvents. Structurally, they belong to five different groups (Figure 1): okadaic acid, including okadaic acid (OA), and dinophysistoxins (DTXs); azaspiracids (AZAs); pectenotoxins (PTXs); yessotoxins (YTXs); and cyclic imines (CIs), which include spirolides (SPXs), pinnatoxins (PnTXs), pteriatoxins (PtTXs), and gymnodimines (GYMs).

Currently, some of these lipophilic toxins are regulated in the European Union. According to the current regulation [1,2], the live bivalve mollusks placed on the market for human consumption must not contain marine biotoxins in total quantities (measured in the whole body or any edible part separately) that exceed: OA, DTXs and PTXs together, 160 µg of okadaic acid equivalents per kilogram; YTXs, 3.75 mg of yessotoxin equivalent per kilogram; and AZAs, 160 micrograms of azaspiracid equivalents per kilogram. On the contrary, no limits are established for any compounds belonging to the CI group [3]. However, several clues indicate that it is more than likely that these “emerging” toxins will have to be considered in the not too distant future: Fundamentally, their increased occurrence in shellfish all over the world [4,5,6,7,8,9,10,11,12], probably due to climate change which is affecting the distribution of their producing phytoplankton species [3,13,14].

For effective monitoring programs, efficient, fast, and low-cost methods are crucial. The high sensitivity and selectivity of liquid chromatography-mass spectrometry make this technique the best choice for the simultaneous evaluation of different lipophilic toxins [16]. In the current European Union regulation [17], liquid chromatography coupled to tandem mass spectrometry (LC-MS/MS) has been established as the reference methodology for monitoring the regulated lipophilic toxins. Because of this, the European Reference Laboratory in Marine Biotoxins (EURLMB) published an EU-Harmonized Standard Operating Procedure (SOP) [18] for the determination of lipophilic marine biotoxins in mollusks. This SOP only includes the regulated toxins and proposes six different chromatographic methods being the fast one 9 minutes long. In recent years, numerous LC-MS /MS methods for the detection of the regulated lipophilic toxins together with some cyclic imines have been published [19,20,21,22,23]. However, the run time of the required chromatograms is relatively long, and they do not include, at least, some especially important cyclic imines such as 13 desmethylspirolide C (13desmSPXC), gymnodimine A (GYMA), 13,19 didesmethylspirolide C (13,19didesmSPXC), 20 methylspirolide G (20MethylSPXG), or pinnatoxin G (PnTXG).

For the Galician monitoring program, Intecmar adapted and accredited (Accreditation N° 160/LE 394) an LC-MS/MS method considering the guidelines of the EURLMB SOP [18] and based on the methods described by Gerssen et al. [20] and Regueiro et al. [24]. The method uses a chromatographic column with sub-2 µm particles and allows for identifying and quantifying the regulated lipophilic toxins in a 9-minute run. The development of the core-shell or fused-core silica particles technology for the chromatographic columns allows for efficient chromatographic separations avoiding some of the drawbacks of the sub-2 µm technology [25], as high backpressure and clogging problems. These characteristics are especially interesting for monitoring systems because: (a) working with lower pressures reduces the stress of the chromatographic equipment making it them less prone to failures; and (b) using larger particles reduces the risk of aggregation of compounds in the column head which can interfere the analyses during long chromatographic batches.

This work describes the validation of a fast LC-MS/MS method, using a fused-core technology chromatographic column, for the determination, in a 3.6 minutes run, of fourteen lipophilic toxins (OA, DTX1, DTX2, PTX2, YTX, HomoYTX, AZA1, AZA2, AZA3, 13desmSPXC, PnTXG, GYMA, 13,19didesmSPXC and 20Methyl SPXG) in bivalve mollusks.

## 2. Results and Discussion

The LC-MS/MS method developed allows for, as far as we know for the first time, the determination of fifteen lipophilic toxins (Figure 2) (fourteen validated) in a single chromatographic run of 3.6 min. All toxins elute in the first 2.3 min of chromatogram. The peaks have good shape and resolution allowing for correct identification and quantification of the compounds. The 45-OH YTX was not included in the validation because no reference material is commercially available. Using 2.6 µm core-shell (present work) instead of sub-2 µm [22,23] particles allows for effectively reducing the analysis time because it is possible to increase the flow rate maintaining the backpressure, with only marginal losses in efficiency and resolution. Therefore, the analysis time per sample of the developed method is substantially reduced in comparison to previously proposed method [18,21,22,23,24,26,27].

### 2.1. LOQ/LOD

The limits of detection (LOD) ranged from 0.035 µg kg^−1^ to 24 µg kg^−1^ and the limits of quantification (LOQ) ranged from 0.12 µg kg^−1^ to 74 µg kg^−1^ (Table 1). The lowest and the highest values were for 13,19 didesmSPXC and HomoYTX, respectively.

Even injecting only 1 µL of sample, LOD and LOQ values are notably below the concentrations reported with other equipment/method, probably because a highly sensitive mass spectrometer was used. In this sense, there is a wide variety of LOQ and LOD data published for lipophilic toxin analyses using Agilent [22,23,26], Thermo [24,27] or Waters [20,21] mass spectrometers among others. Almost all of these papers reported LOQs and LODs values higher than those in the present work, except O‘Neill et al. [28] and Yang et al. [27] with similar and slightly lower values, respectively. However, none of them combines the 14 toxins in the same run and in such a short time

The LOQs values obtained were checked experimentally by fortifying the methanolic extracts of three blank samples (mussel, clam, and oyster) at concentrations close to the theoretical LOQ of each toxin (see selectivity/specificity section).

### 2.2. Selectivity/Specificity

The analyses to evaluate the recovery of fortified samples at levels around LOQ have been performed under repeatability conditions (*n* = 3, one day). Recoveries obtained for the three matrices were, in general, good (Table 2). On average for the analysed compounds, they ranged from 84.8 to 111.1%, except for 13,19 didesmSPXC (79.9 %), PnTXG (64.4%), and YTX (123.1 %). The recovery for YTX, was over 100 % for mussels (135.5%) and oysters (137.2%) but not for clams (96.7%). For PnTXG, it was below 100% for mussels (56.9 % of recovery) and oysters (57.7%). The high (for YTXs) and low (for PnTXG) recoveries for mussels and oysters, were the reason why the RSD_r_ (species averaged) of these two toxins were close to 20% instead 15.3% as in the other studied toxins (Table 2). The high value obtained for 13desmSPXC in oysters (142.9%) was due to the presence of traces of this toxin in the blank sample, so it was not included in the computations of mean, SD and RSD_r_. The low recovery of GYMA in the oyster sample (42.4%) should attributed to its concentration after spiking, that was below its LOQ. This value has not been included in the mean, SD and RSD_r_ calculations.

Since it was the extract that were spiked with toxins, the cause of the differences in recovery should be matrix effect. Matrix effect (ion suppression or ion enhancement) in LC-MS/MS analysis is well known, and can lead to under or overestimate toxin concentration. It is heavily dependent on the analytical system used [29] and, although different strategies have been used to minimize it [24,30] (e.g., SPE clean-up [31]), it is, still present in many published methods [22,23,26].

### 2.3. Linearity

The method has a good linearity. A straight line fitted well the calibration curve of each toxin, with r^2^ always above 0.989 (Table 3). Relative slope values were always between 80 and 120%. The random distribution of the residuals around the calibration line confirmed the linearity of these intervals.

### 2.4. Precision: Repeatability and Reproducibility

The obtained RSD_r_ and RSD_R_ values ranged from 3.1 to 21.9% and 3.0 to 17.4% respectively (Table 4) (except for Level 1 of GYMA and 13desmSPXC). These values comply with IUPAC, FDA and SANCO guidelines (15–20%) confirming the good precision of the method. The high standard relative deviations obtained in Level 1 for 20 MethylSPXG (17.8 and 16.5% for RSD_r_ and RSD_R_ respectively) and GYMA (36 and 30.9% for RSD_r_ and RSD_R_) are not indicative since they were spiked to concentrations below their LOQ. RSD_r_ and RSD_R_ measured for 13desmSPXC in oysters at Level 1 were abnormally high due to the presence of traces of this toxin in the sample used as blank (it was not possible to obtain oyster completely free of this toxin), these data have not been included in the general computation. All other Level 1 data of this 13desmSPXC, as well as those of PnTXG, were also high (13desmSPXC: 21.2 and 30.6%, PnTXG: 21.9 and 14.8%, for RSD_r_ and RSD_R_ respectively). RSD_r_ and RSD_R_ obtained for OA, PTX2, AZA1, YTX were better than others previously published, for example, by Villar-González et al. [32].

Available naturally contaminated samples used for the precision study only contained OA and derivatives, so the analyses were carried out after performing alkaline hydrolysis (see material and methods section). The absence of natural samples contaminated with toxins other than OA, as it happens in most occasions in Galicia [33], made it impossible to carry out precision studies for components other than that compound. The precision, therefore, was evaluated using the estimations of total OA (measured as free OA after alkaline hydrolysis). For all species, the RSD_r_ (repeatability conditions) were below 11.3%. The maximum RSD_R_ was 14.5% in a mussel sample. Although all RSD_R_ were acceptable, the values obtained in mussels were be slightly higher than in other species (Table 5). Additional information about precision in naturally contaminated samples is provided in Appendix A.

### 2.5. Accuracy

Freeze-dried mussel (*Mytilus edulis*) tissue certified reference material (CRM -FDMT) has been analyzed seven times on two different days. Except for YTX, all the estimated values were between 92.4 and 132% of the certified concentration (recoveries) (Table 6). Large deviations have been observed for YTX with values of 154.1% the certified concentrations.

No commercially available certified reference materials have been found in matrices other than mussel. Therefore, to complete the studies, blank bivalve matrices (fortified at four or five concentration levels with certified solutions of lipophilic toxins were prepared (see material and methods section, point 3.6.4. for levels)). For each fortified level, recoveries have been calculated using the mean values obtained from three replicates analyzed in three different days in the three chosen blank matrices (*n* = 27) (Table 7). In the case of the regulated toxins, recoveries were good, always above 84% at the four evaluated levels. For the emerging toxins evaluated (13desmSPXC, 13,19didesmSPXC, 20MethylSPXG, GYMA and PnTXG) recoveries were also acceptable, ranging from 71.1 to 100.8%, except for PnTXG and 13,19didesmSPXC which were 62% and 67.1% respectively (Table 7). As already explained, 13desmSPXC data from oysters at Level 1 have not been considered, since the value was abnormally high, due to the presence of toxin traces in the blank sample. The relatively low recoveries obtained for PnTXG (62–77.1%) and 13,19didesmSPXC (67.1–73.7%) could be attributed to the different matrices used, being the recoveries obtained in clams higher than those in oyster and mussel.

## 3. Materials and Methods

### 3.1. Chemicals

Acetonitrile (LC-MS grade) and methanol (HPLC grade quality) were purchased from Scharlab (Spain) and VWR (Spain), respectively. Ultrapure water was obtained from a Milli-Q gradient system fed with an Elix Advantage-10 (Millipore Ibérica, Spain). Ammonium hydroxide (NH_4_OH, 25%) and sodium hydroxide (NaOH > 99%) were obtained from Merck (Barcelona, Spain), and hydrochloric acid (HCl, 37%) from Panreac (Barcelona, Spain).

### 3.2. Reference Materials

Certified reference standards (CRMs) for DTX1, YTX, HomoYTX, AZA1, AZA2 and AZA3, and quality control standards (QCSs) for 13desmSPXC, 13,19didesmSPXC and 20MethylSPXG were obtained from CIFGA, S.A. (Lugo, Spain). OA, DTX2, PTX2, GYMA and PnTXG certified solutions and the certified reference material CRM-FDMT1 (freeze-dried mussel tissue) containing OA, DTX2, DTX1, PTX2, 13desmSPXC, YTX, AZA1, AZA2 and AZA3, were acquired from the Institute for Marine Biosciences, National Research Council (NRC), Halifax, Nova Scotia, NS, Canada.

### 3.3. Shellfish Samples

Method validation has been performed with whole flesh of mussels (*Mytilus galloprovincialis*), clams (*Ruditapes philippinarum*) and oysters (*Ostrea edulis*). All of them were collected from shellfish harvesting areas of Galicia and fortified with certified reference solutions. Furthermore, naturally contaminated samples of mussels (*M. galloprovincialis*), clams (*R. philippinarum*), cockles (*Cerastoderma edule*) and razor clams (*Ensis siliqua*) from the Galician Rías were also analyzed.

### 3.4. Sample Extraction and Hydrolysis

Approximately 100 g of soft tissues of each tested bivalve (mussels, clams and oysters), not containing toxins (blank), were homogenized. For OA, DTXs, PTX2 and AZAs, aliquots of these blank samples were spiked with certified reference solutions to concentrations equivalent to LOQ, 0.5, 1, and 1.5 times the regulated levels (see point 3.6.4. for levels). For YTX and HomoYTX the concentrations were lower because their very high regulation limit makes it impossible to attain the required concentration without applying some concentration step to the currently available reference material (see point 3.6.4. for levels). Emerging toxins (13desmSPXC, GYMA, 13,19didesmSPXC, 20MethylSPXG and PnTXG) do not have regulated limits in the EU legislation so, fortification levels have been arbitrary established (see point 3.6.4. for levels). The freeze-dried reference material CRM-FDMT has been reconstituted following the procedure recommended by the manufacturer (NRC-CNRC). Briefly, CRM-FDMT (0.35 g) was reconstituted in a 50 mL centrifuge tube by adding 1.65 mL deionized water, followed by vortex mixing for 30 s and sonication for 1 min in an ultrasonic bath.

Raw, fortified samples and the CRM-FDMT material were extracted according to the standardized operating procedure of the EU-RL for the determination of marine lipophilic biotoxins in mollusks [18]. Briefly, for the analysis of free forms of the OA group (OA, DTXs) and most other lipophilic toxins (PTX2, 13desmSPXC, YTX, HomoYTX, AZA1, AZA2, AZA3, GYMA, 13,19didesmSPXC, 20MethylSPXG, PnTXG) 2-gram aliquots of homogenized tissues were extracted with 9 mL of MeOH 100% (twice) and centrifuged at 2000× *g* for 10 min. Both supernatants were combined and the final volume was adjusted to 20 mL. An aliquot was filtered through a 0.22 µm PVDF syringe filter, diluted 5/8 with methanol and analyzed by LC-MS/MS. Total OA concentration (main toxin plus its hydrolysable derivatives) was determined after alkaline hydrolysis. For it, 625 µL of 2.5M NaOH were added to a 5 mL aliquot of the methanolic extract, vortexed for 30 s and heated at 76 °C for 40 min; time elapsed, the hydrolysate was cooled to room temperature, weighed to check for solvent losses and neutralized by adding the same volume of 2.5M HCl. As for the analysis of the free forms, an aliquot was filtered through a 0.22 µm syringe filter and diluted with MeOH.

### 3.5. LC–MS/MS Procedure

The analyses have been carried out on an Exion LC AD™ System (SCIEX, Framingham, MA, USA) coupled to a Qtrap 6500+ mass spectrometer (SCIEX) through an IonDrive Turbo V interface in electrospray mode. The toxins were separated in a Phenomenex Kinetex EVO C18 “core-shell” column 50 mm (length) × 2.1 mm (id), 2.6 µm (particle size). Mobile Phase A was water and B MeCN 90%, both containing 6.7 mM NH_4_OH (pH 11) [20]. The gradient started with 22% B, was maintained for 0.1 min, followed by a linear increment to reach 95% B at minute 1.8, and maintaining this composition until minute 2.90. The composition was then returned linearly to the initial one in 0.20 min and maintained 0.5 min before the next injection. The flow rate was 1,000 µLmin^−1^, the injection volume was 1 µL and the column temperature was 40 °C.

The mass spectrometer parameters were optimized by direct infusion using toxin standards, when available, and were set to: ion source Gas 1, 75 (arbitrary units); ion source Gas 2, 75 (arbitrary units); ion spray voltage, 5000 (positive) and −4500 (negative); capillary temperature, 600 (°C); curtain gas, 30; collision Gas, medium. Specific MS/MS fragmentation conditions and collision energies for all the toxins validated are shown in Table 8. The transition with the highest intensity (qn) was used as for quantification, while the second transition (ql, the one with the lowest intensity) was used for confirmatory purposes.

### 3.6. Scope and Method Validation

The scope of the method includes lipophilic regulated toxins: OA, DTX2, DTX1, PTX2, YTX, HomoYTX, AZA1, AZA2, and AZA3 and five emerging unregulated toxins: 13desmSPXC, GYMA, 13,19diDesMethylSPXC, 20MethylSPXG and PnTXG.

The method has been validated according to the International Organization for Standardization, International Union of Pure and Applied Chemistry (IUPAC), and following the AOAC and Eurachem guidelines. The following parameters have been assessed: LOQ/LOD, specificity, linearity, precision (repeatability and reproducibility) and accuracy.

#### 3.6.1. LOQ/LOD

For the theoretical estimation of LOQ and LOD, a standard curve of toxins has been prepared and expanded with dilutions (2, 4, 6 and 10×) of Level 1 (the one with the lowest concentration). Signal-to-noise ratio (S/N) has been determined directly in each chromatogram for the second transition (the one with the lowest intensity) with Analyst software (SCIEX). Theoretical LOQs for each toxin have been calculated as 10 times the S/N ratio. Theoretical LODs have been estimated by dividing the LOQ by 3.33 to attain an S/N of 3.

#### 3.6.2. Selectivity/Specificity

The selectivity/specificity test was conducted by comparing chromatograms of three blank matrices (*M. galloprovincialis*, *R. philippinarum* and *O. edulis*) with their corresponding methanol extracts fortified with reference solutions at a level close to the theoretical LOQ of each of the toxins evaluated (Table 1). For emerging toxins (13,19didesmSPXC, 20MethylSPXG, GYMA and PnTXG) and to simplify the experimental design, the reference solution used for fortifying samples contained the same toxin concentrations for all of them (3 µg kg^−1^). For this reason, in the case of GYMA and 20MethylSPXG fortified values are slightly lower than their LOQ and for 13,19didesmSPXC and PnTXG are higher to it. Recoveries (%), mean, standard deviation (SD) and relative standard deviation of repeatability (RSD_r_ %) of the three fortified species have been calculated for each toxin.

#### 3.6.3. Linearity

In this study, linearity has been estimated using a calibration solution curves prepared by dilution CRMs in methanol, and analyzed in triplicate. The concentration levels used are those in Table 9. The calibration line of each toxin and its corresponding coefficient of determination (r^2^), were calculated by least squares regression weighed with 1/x. Linearity evaluation has been carried out by calculating how many points deviate from the calibration curve according to the next formula: relative slope = (Y_i_/X_i_) × 100 / ∑ (Y_i_/X_i_) / N, where: X_i_: toxin concentration (ng mL^−1^); Y_i_: area of the peak, and N: number of pairs of Y_i_ / X_i_ values. The acceptance range was defined as 80–120%.

Residual values have also been calculated, plotted and visually examined to detect suspicious deviations or curvature of the response.

#### 3.6.4. Precision: Repeatability and Reproducibility

Precision has been evaluated using fortified methanolic extracts and naturally contaminated samples. Since precision frequently varies with concentration of analyte, it has been evaluated at different concentrations. The extracts of mussels, clams, and oysters from Galicia, were spiked at the four (or five for 13desmSPXC) concentration levels indicated in the Table 10. Three replicates of each fortified sample were analyzed under repeatability conditions (same day), and intra-laboratory reproducibility conditions (same equipment, same laboratory, three different days).

Contaminated natural samples consisted of mussels (*M. galloprovincialis*), cockles (*C. edule*), clams (*R. philippinarum*) and razor clams (*E. siliqua*) from Galician Rías, containing OA. They have been analyzed five times under repeatability conditions in one day and five different days for reproducibility. The concentration estimates have been compared to those obtained with other method and LC-MS/MS (Xevo TQ-S triple quadrupole mass, Waters) according to the accredited method (Accreditation N°160//LE 394) used for monitoring by Intecmar and based on Gerssen et al. [20] and Regueiro et al. [24]. After the analyses, mean concentration, SD, RSD_r_ (for repeatability, 1 day), and RSD_R_ (for reproducibility, 5 days) have been calculated.

#### 3.6.5. Accuracy

A freeze-dried mussel tissue certified reference material for multiple marine toxins (CRM-FDMT1) has been used to evaluate the accuracy and traceability of the method. Eight replicates of a methanolic extract on two different days were analyzed in order to estimate the combined effects of the method and the particular laboratory bias. The recovery percentages and/or the bias of the estimates (in relation to the reference value) have been calculated.

## 4. Conclusions

The performance of the method developed in this work makes it suitable for accurate measurements of the lipophilic toxins regulated by EU legislation (OA, DTX2, DTX1, PTX2, YTX, HomoYTX, AZA1, AZA2, AZA3), and at least five emerging toxins (13desmSPXC, GYMA, 13,19didesmSPXC, 20MethylSPXG, PnTXG). The method allows for the detection and quantification of these fourteen toxins in 3.6 minutes with high specificity, robustness, and with LOQ and LOD low enough to ensure food safety. As far as we know, this method is the fastest one that allows the quantification of so many lipophilic toxins in a single run.

## Figures and Tables

**Figure 1 marinedrugs-19-00603-f001:**
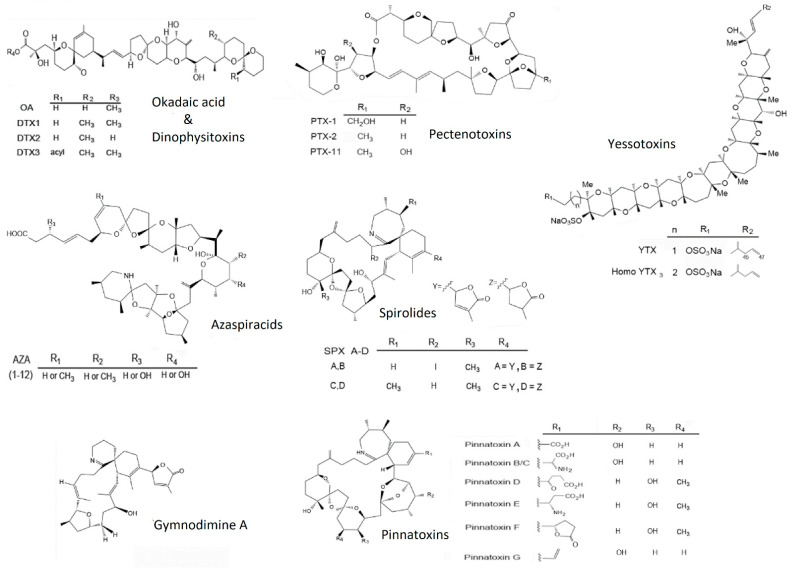
Structure of the main lipophilic toxins. Source: Leyva-Valencia et al. [15].

**Figure 2 marinedrugs-19-00603-f002:**
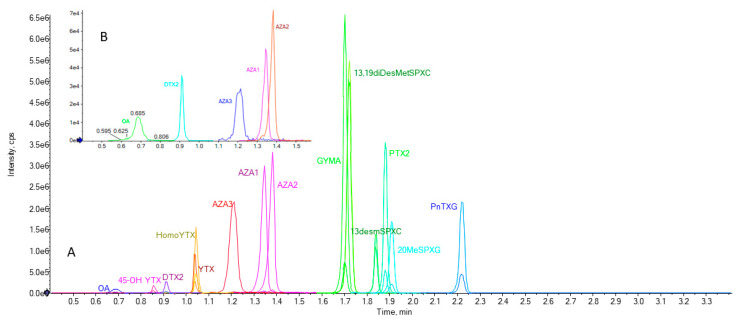
Chromatogram with two MRM transitions (quantifier and qualifier) obtained from a mixture of reference solutions. Abbreviations: OA (okadaic acid), 45-OH YTX (45 hydroxyyessotoxin), DTX2 (dinophysistoxin 2), HomoYTX (Homoyessotoxin), YTX (yessotoxin), DTX1 (dinophysistoxin 1), AZA3 (azaspiracid 3), AZA1 (azaspiracid 1), AZA2 (azaspiracid 2), GYMA (gymnodimine A), 13,19diDesMetSPXC (13,19 didesmethylspirolide C), 13desmSPXC (13 desmethylspirolide C), PTX2 (pectenotoxin 2), 20MeSPXG (20 methylspirolide G), PnTXG (pinnatoxin G) (**A**); Zoom of second MRM transitions (qualifier) for OA, DTX2 and AZAs toxins (**B**).

**Table 1 marinedrugs-19-00603-t001:** Limits of detection (LOD) and quantitation (LOQ) obtained for fourteen lipophilic toxins.

Toxin	LOD (µg kg^−1^)	LOQ (µg kg^−1^)
OA	3	10
DTX2	9	29
DTX1	9	29
AZA1	9	29
AZA2	9	29
AZA3	9	29
YTX	22	72
HomoYTX	24	74
PTX2	7.3	25
13desmSPXC	0.2	0.67
13,19didesmSPXC	0.035	0.12
20MethylSPXG	1.1	3.8
GYMA	1.2	3.9
PnTXG	0.4	1.3

**Table 2 marinedrugs-19-00603-t002:** Recovery values in fortified samples at levels around LOQ and performed under repeatability conditions.

	OA	DTX2	DTX1	AZA1	AZA2	AZA3	PTX2	YTX	HomoYTX	13desmSPXC	13,19didesmSPXC	20MethySPXG	GYMA	PnTXG
Fortified levels (µg kg^−1^)														
	30 *	30	30	30	30	30	30	75	75	0.7	3 *	3 **	3 **	3 *
Matrix	Recovery (%)													
Mussel	89.3	100.3	114.7	89.7	82.3	87.3	94.3	135.5	119.2	100	84.3	105.7	103.3	56.9
Clam	71	80.7	96.7	75.7	91.7	79.3	81.3	96.7	92	85.7	79.7	99.3	101.7	78.6
Oyster	94	105.7	122	103	99.7	95	100.3	137.2	111.1	142.9 ^a^	75.7	91.7	42.4 ^a^	57.7
Mean (*n* = 3)	84.8	95.6	111.1	89.5	91.2	87.2	92.0	123.1	107.4	92.9 (*n* = 2)	79.9	98.9	102.5 (*n* = 2)	64.4
SD (*n* = 3)	12.2	13.2	13.0	13.7	8.7	7.9	9.7	22.9	14.0	10.1 (*n* = 2)	4.30	7.0	1.1 (*n* = 2)	12.3
RSD_r_, %	14.3	13.8	11.7	15.3	9.6	9	10.6	18.6	13.0	10.9	5.4	7.1	1.1	19.1

* Level of fortified slightly above its LOQ for simplicity of analysis. ** Level of fortified slightly below its LOQ for simplicity of analysis. ^a^ Data not considered for mean, SD and RSD_r_ calculations.

**Table 3 marinedrugs-19-00603-t003:** Coefficient of determination (r^2^) obtained for the fourteen lipophilic toxins.

Toxin	r^2^
OA	0.9922
DTX2	0.9917
DTX1	0.9918
AZA1	0.9972
AZA2	0.9972
AZA3	0.9966
YTX	0.9891
HomoYTX	0.9922
PTX2	0.9969
13desmSPXC	0.9969
13,19didesmSPXC	0.9921
20MethylSPXG	0.9965
GYMA	0.9970
PnTXG	0.9958

**Table 4 marinedrugs-19-00603-t004:** Average RSD_r_ (repeatability) and RSD_R_ (reproducibility) of the studied toxin in the three evaluated matrices.

	Level 1		Level 2		Level 3		Level 4		Level 5	
	RSD_r_, % (*n* = 9)	RSD_R_, % (*n* = 27)	RSD_r_, % (*n* = 9)	RSD_R_, % (*n* = 27)	RSD_r_, %(*n* = 9)	RSD_R_, % (*n* = 27)	RSD_r_, % (*n* = 9)	RSD_R_, % (*n* = 27)	RSD_r_, % (*n* = 9)	RSD_R_, % (*n* = 27)
OA	8.6	6.5	4.4	7.9	4.8	5.8	3.6	5.0		
DTX2	8.5	7.9	4.3	6.4	4.7	6.7	5.7	6.7		
DTX1	10.1	10.0	6.6	9.6	5.6	9.7	3.8	8.9		
AZA1	11.3	10.7	3.1	4.1	3.7	3.2	3.6	3.4		
AZA2	8.1	7.7	4.7	4.9	3.7	3.0	4.1	3.2		
AZA3	9.3	8.5	4.5	4.8	3.0	2.7	4.0	3.2		
YTX	16.0	14.4	14.9	14.2	14.0	13.7	12.5	13.9		
HomoYTX	13.0	11.1	14.0	13.6	12.8	12.7	12.9	13.9		
PTX2	8.1	8.5	9.3	7.2	7.5	5.1	5.8	5.0		
13desmSPXC	21.2 (*n* = 6) *	30.6 (*n* = 18) *	18.3	17.4	6.4	7.4	6.9	6.7	8.9	8.9
13,19didesmSPXC	15.2	14.8	4.9	10.2	5.4	6.4	7.6	8.3		
20MethylSPXG	17.8 **	16.5 **	4.1	4.8	4.7	6.0	6.1	7.5		
GYMA	36.0 **	30.9 **	7.4	7.2	5.9	5.4	7.0	7.3		
PnTXG	21.9	14.8	4.9	5.1	6.3	7.0	7.5	8.7		

* Data from oyster were not considered for RSD calculations. ** Level of fortified slightly below its LOQ for simplicity of analysis.

**Table 5 marinedrugs-19-00603-t005:** Precision results for the analysis of naturally contaminated samples.

Sample	Matrix	Toxin	RSD_r_ % (*n* = 5)	RSD_R_ % (*n* = 10)
508	Mussel	Total OA	10.1	11.7
509	Mussel	Total OA	8.6	14.5
510	Mussel	Total OA	7.9	13.0
512	Mussel	Total OA	9.7	10.3
513	Mussel	Total OA	8.9	12.5
1532	Cockle	Total OA	9.6	12.6 *
1534	Cockle	Total OA	8.7	8.1 *
1540	Razor clam	Total OA	11.2	10.0 *
1541	Cockle	Total OA	11.3	8.7 *
1547	Clam	Total OA	8.0	6.9 *

* *n* = 12

**Table 6 marinedrugs-19-00603-t006:** Recovery and reproducibility for CRM-FDMT material.

Type of Toxin	Mean (µg kg^−1^)	SD (µg kg^−1^)	RSD_R_, % (*n* = 8)	Certified Value	Recovery %
OA	306.7	22.8	7.4	278.3	110.2
DTX2	795.1	64.6	8.1	624.8	129.1
DTX1	155.7	11.6	7.5	119.0	132.0
AZA1	809.1	70.5	8.7	717.5	112.3
AZA2	194.3	12.9	6.6	197.8	98.1
AZA3	186.9	12.5	6.7	168	110.6
YTX	670.0	176.5	26.3	435.8	154.1
PTX2	107.6	12.7	11.8	115.5	92.4
13desmSPXC	509.0	35.4	6.9	472.5	108.0

**Table 7 marinedrugs-19-00603-t007:** Toxin recoveries for different fortification levels in the three matrices studied under reproducibility conditions.

	Recovery, % (Mean ± SD), *n* = 27				
	Level 1	Level 2	Level 3	Level 4	Level 5
OA	84.8 ± 1.6	85.6 ± 5.4	91.7 ± 8.5	93.8 ± 14.9	
DTX2	90.4 ± 2.1	92.5 ± 4.7	99.6 ± 10.6	101.0 ± 21.6	
DTX1	100.7 ± 3.0	105.7 ± 8.2	110.7 ± 17.2	112.9 ± 32.3	
AZA1	89.9 ± 2.9	91.2 ± 3.0	96.0 ± 4.9	98.7 ± 10.7	
AZA2	90.8 ± 2.1	91.8 ± 3.4	97.0 ± 4.6	99.3 ± 10.0	
AZA3	88.2 ± 2.3	91.0 ± 3.5	97.4 ± 4.20	99.8 ± 10.3	
YTX	107.9 ± 11.6	113.0 ± 32.2	121.1 ± 66.6	120.5 ± 134.2	
HomoYTX	96.2 ± 8.0	100.7 ± 27.3	107.8 ± 54.7	107.4 ± 119.0	
PTX2	85.7 ± 2.1	84.9 ± 4.9	89.1 ± 7.2	91. 5 ± 14.6	
13desmSPXC	100.8 ± 0.2 (*n* = 18) *	71.1 ± 1.0	76.5 ± 1.3	83.4 ± 2.5	87.5 ± 6.9
13,19didesmSPXC	72.8 ± 0.3	71.8 ± 2.2	67.1 ± 3.2	73.7 ± 9.2	
20MethylSPXG	92.8 ** ± 0.5	98.9 ± 1.4	92.3 ± 4.1	101.7 ± 11.5	
GYMA	83.5 ** ± 0.8	97.2 ± 2.1	90.6 ± 3.7	99.3 ± 10.8	
PnTXG	62.0 ± 0.3	75.8 ± 1.2	70.3 ± 3.7	77.1 ± 10.0	

* Data from oyster have not been considered for recovery calculations. ** Level of fortified slightly below its LOQ for simplicity of analysis.

**Table 8 marinedrugs-19-00603-t008:** MS/MS fragmentation conditions for lipophilic toxin determination. ESI = electrospray ionization mode, Q1 = *m*/*z* ratio in the first quadrupole, Q3 = *m*/*z* ratio in the third quadrupole, DEP(v) = declustering potential, EP(v) = entrance potential, CE(v) = collision energy, and CXP(v) = collision cell exit potential, qn = ion pair for quantitation and ql = ion pair for qualifier purpose.

Toxin	ESI	Q1	Q3	DEP (v)	EP (v)	CE (v)	CXP(v)
OA_DTX2 (qn)	NEG	803.52	255.15	−80	−15	−62	−11
OA_DTX2 (ql)	NEG	803.52	563.40	−80	−15	−60	−11
YTX (qn)	NEG	570.43	467.40	−80	−15	−42	−11
YTX (ql)	NEG	570.43	396.40	−80	−15	−42	−11
HomoYTX (qn)	NEG	577.40	474.40	−80	−15	−42	−11
HomoYTX (ql)	NEG	577.40	403.40	−80	−15	−42	−11
DTX1 (qn)	NEG	817.50	255.15	−80	−15	−60	−11
DTX1 (ql)	NEG	817.50	563.45	−80	−15	−52	−11
AZA3 (qn)	POS	828.46	810.5	80	15	30	10
AZA3 (ql)	POS	828.46	658.4	80	15	43	10
AZA1 (qn)	POS	842.46	824.5	80	15	30	10
AZA1 (ql)	POS	842.46	672.4	80	15	43	10
AZA2 (qn)	POS	856.46	838.5	80	15	30	10
AZA2 (ql)	POS	856.46	672.4	80	15	43	10
GYMA (qn)	POS	508.33	490.2	80	15	50	10
GYMA (ql)	POS	508.33	136.00	80	15	50	10
13,19didesmSPXC (qn)	POS	678.5	164.00	80	15	60	10
13,19didesmSPXC (ql)	POS	678.50	430.3	80	15	50	10
13desmSPXC (qn)	POS	692.5	164.3	80	15	60	10
13desmSPXC (ql)	POS	692.50	444.3	80	15	55	10
PTX2 (qn)	POS	876.46	823.5	80	15	35	10
PTX2 (ql)	POS	876.46	213.1	80	15	50	10
20MethylSPXG (qn)	POS	706.5	164.3	80	15	60	10
20MethylSPXG (ql)	POS	706.50	348.3	80	15	55	10
PnTXG (qn)	POS	694.5	164.3	80	15	60	10
PnTXG (ql)	POS	694.5	440.3	80	15	50	10

**Table 9 marinedrugs-19-00603-t009:** Toxin concentrations in ng mL^−1^ used to evaluate the linearity of the method.

OA, DTXs, AZAs, PTX2	YTX, HomoYTX	13desmSPXC	13,19didesmSPXC, 20MethylSPXG, GYMA, PnTXG
2.1	5.2	0.19	0.47
6.3	15.7	0.6	0.9
10.6	26.5	1.8	1.9
16.9	42.1	3.0	3.75
21.6	53.9	4.7	7.5
31.7	79.2	6	15
45.7	114.29	8.8	
		12.7	

**Table 10 marinedrugs-19-00603-t010:** Toxin concentrations in µg kg^−1^ used to fortify blank samples.

	Level 1	Level 2	Level 3	Level 4	Level 5
OA	30	80	160	320	
DTXs	30	80	160	320	
AZAs	30	80	160	320	
PTX2	30	80	160	320	
YTX	75	200	400	800	
HomoYTX	75	200	400	800	
13desmSPXC	0.7	8.3	22.2	44.4	89
GYMA	3	30	75	150	
PnTXG	3	30	75	150	
13,19didesmSPXC	3	30	75	150	
20MethylSPXG	3	30	75	150	

## Data Availability

Data are available on request in the Centro de Investigacións Mariñas.

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
