# Peer review of "Development of a Fast Liquid Chromatography Coupled to Mass Spectrometry Method (LC-MS/MS) to Determine Fourteen Lipophilic Shellfish Toxins Based on Fused–Core Technology: In-House Validation"

_marinedrugs, 2021, doi:10.3390/md19110603_

Round 1

Reviewer 1 Report

The development of quick and effective method for the analysis of marine toxin is very important for consumer food safety protection. The authors had tried to develop a fast and comprehensive LC-MS method for regulated and emerging toxins. But the current work is not well-organized, and the authors failed to give a good description of their work. This work is not easy to understand. The following improvement should be made:

  1. The text description of the table could not match many times in the current work. For example, there are no Table 1 and Table 2 in the current work.
  2. Page 3, Figure 1 were obtained from mixture of reference solution level 6 of the calibration curves. Throughout the whole manuscript, the authors did not provide information of the level 6 calibration solutions.
  3. Page 3, Line 106-107. The authors mentioned most of the reported LOD and LOQ by different instruments like Agilent, Thermo, and Waters were higher than their current work. The performance and sensitivity of different LC-MS/MS spectrometer could be widely different. The authors should also compare the LOD and LOQ results obtained from instrument with similar sensitivity or performance as SCIEX Triple Quad 6500+ mass spectrometer.
  4. Page 4, 2.2 Selectivity/Specificity. The authors mentioned “In the present study, the analyses to evaluate the recovery of fortified samples at LOQ levels were performed under repeatability conditions (n=3, one day).”. For 13,19didesmSPXC and PnTXG, the LOQ were 0.12 and 1.3 mg/kg respectively. The spiked levels in Table 4 for 13,19didesmSPXC and PnTXG were both 3 mg/kg which were not the same as LOQ. The authors should explain the difference for clarity.
  5. Page 4, Line 128-130, the authors mentioned positive matrix effect for YTX and contrary to previous report of negative effect. Please explain the difference for YTX recovery.
  6. Page 5, Section 2.3 Linearity, Line 150-158, no detail data for linearity is shown for all the analyzed toxins. This should be listed in one table.
  7. Page 5, Table 5, RSDr and RSDR were done on 9 and 27 replicates, respectively. The authors should explain how they adopt their validation schemes for the number of replicates.
  8. Page 6, Table 6, RSDr and RSDR were done on 2 and 5 replicates, respectively. The authors should explain how they adopt their validation schemes for the number of replicates.
  9. Page 8, Line 227, “table 9” should be Table 8.
  10. Page 10, Line 327, Table 1 should be Table 9.
  11. Page 10, Table 9, the authors should distinguish the ion pairs for qualifier and quantitation purpose.
  12. Page 11, Table 9. PnTXG only one ion pair for MRM is specified.
  13. Page 11, Line 348-349. The authors should give better description about “the second transition (the one with the lowest intensity)” and mark them in Table 9.
  14. Extracted ion chromatograms of the selected MRM transitions for all the toxins analyzed should be included.
  15. Page 11. Section3.6.2 Selectivity/Specificity is extra content and duplicate of Section 2.2 Selectivity/Specificity in Page 4. The authors could provide other information such as ion ratio and retention time shift.
  16. Page 12, Line 384-385. Both RSDr and RSDR were done on 2 replicates, this need to be carefully checked whether is meaningful enough for method validation. 

Author Response

Reviewer 1

The development of quick and effective method for the analysis of marine toxin is very important for consumer food safety protection. The authors had tried to develop a fast and comprehensive LC-MS method for regulated and emerging toxins. But the current work is not well-organized, and the authors failed to give a good description of their work. This work is not easy to understand. The following improvement should be made:

1.The text description of the table could not match many times in the current work. For example, there are no Table 1 and Table 2 in the current work.

The numbering of the tables has been corrected

2. Page 3, Figure 1 were obtained from mixture of reference solution level 6 of the calibration curves. Throughout the whole manuscript, the authors did not provide information of the level 6 calibration solutions.

To know the concentrations of level 6 of calibration does not provide any information to the chromatogram, so we decided to eliminate it to avoid confusion.

3. Page 3, Line 106-107. The authors mentioned most of the reported LOD and LOQ by different instruments like Agilent, Thermo, and Waters were higher than their current work. The performance and sensitivity of different LC-MS/MS spectrometer could be widely different. The authors should also compare the LOD and LOQ results obtained from instrument with similar sensitivity or performance as SCIEX Triple Quad 6500+ mass spectrometer.

As far as we known there are no paper published with an MS with a similar sensitivity. The most similar one is a Waters Xevo TQ-S which the information has been included in the manuscript.

4. Page 4, 2.2 Selectivity/Specificity. The authors mentioned “In the present study, the analyses to evaluate the recovery of fortified samples at LOQ levels were performed under repeatability conditions (n=3, one day).”. For 13,19didesmSPXC and PnTXG, the LOQ were 0.12 and 1.3 mg/kg respectively. The spiked levels in Table 4 for 13,19didesmSPXC and PnTXG were both 3 mg/kg which were not the same as LOQ. The authors should explain the difference for clarity.

An explanation of why for some toxins fortified levels have been made slightly above or below it LOQ has been included in the section 3.6.2:

“For emerging toxins (13,19didesmSPXC, 20MethylSPXG, GYMA and PnTXG) and to simplify the experimental design, the reference solution used for fortifying samples contained the same toxin concentrations for all of them (3 µg kg-1). For this reason, in the case of GYMA and 20MethylSPXG fortified values are slightly lower than their LOQ and for 13,19didesmSPXC and PnTXG are higher to it.”

5. Page 4, Line 128-130, the authors mentioned positive matrix effect for YTX and contrary to previous report of negative effect. Please explain the difference for YTX recovery.

The whole paragraph has been rewritten for a better understanding.

6. Page 5, Section 2.3 Linearity, Line 150-158, no detail data for linearity is shown for all the analyzed toxins. This should be listed in one table.

Table 3 with r2 values has been added

7.Page 5, Table 5, RSDr and RSDR were done on 9 and 27 replicates, respectively. The authors should explain how they adopt their validation schemes for the number of replicates

According to the Eurachem Guide (The Fitness for Purpose of Analytical Methods. A Laboratory Guide to Method Validation and Related Topics. Second Edition 2014) on which the validation carried out is based (point 3.6): “Evaluation of precision requires sufficient replicate measurements to be made on suitable materials. The materials should be representative of test samples in terms of matrix and analyte concentration, homogeneity and stability, but do not need to be CRMs. The replicates should also be independent, i.e. the entire measurement process, including any sample preparation steps, should be repeated. The minimum number of replicates specified varies with different protocols, but is typically between 6 and 15 for each material used in the study”

8. Page 6, Table 6, RSDrand RSDRwere done on 2 and 5 replicates, respectively. The authors should explain how they adopt their validation schemes for the number of replicates.

Following the reviewer's advice, we have been decided to repeat the analysis of 10 samples to study precision by increasing the number of replicates (Table 5). In this way, it increased to n=5 for RSDr and therefore to n=10 or 12 for RSDR. This number of replications is equal (for RSDr) to or even greater (for RSDR) than that employed, for example by Villar-González et al. 2011 in a single- laboratory validation for the determination of lipophilic toxins by LC-MS/MS. Data previously available for other samples have been included as supplementary material (Table S1).

9. Page 8, Line 227, “table 9” should be Table 8.

The numbering of the Tables has been corrected

10. Page 10, Line 327, Table 1 should be Table 9.

The numbering of the Tables has been corrected

11. Page 10, Table 9, the authors should distinguish the ion pairs for qualifier and quantitation purpose.

Ion pairs for qualifier and quantitation purposes have been indicated in the Table 8

12. Page 11, Table 9. PnTXG only one ion pair for MRM is specified.

The second ion pair for PnTXG has been added in Table 8.

13. Page 11, Line 348-349. The authors should give better description about “the second transition (the one with the lowest intensity)” and mark them in Table 9.

Improved description and quantifier and qualifier ion are marked in Table 8.

“The transition with the highest intensity (qn) was used as for quantification, while the second transition (ql, the one with the lowest intensity) was used for confirmatory purposes.”

14. Extracted ion chromatograms of the selected MRM transitions for all the toxins analyzed should be included.

All extracted ion chromatograms have been included in current Figure 2

15. Page 11. Section 3.6.2 Selectivity/Specificity is extra content and duplicate of Section 2.2 Selectivity/Specificity in Page 4. The authors could provide other information such as ion ratio and retention time shift.

Section 3.6.2 does not duplicate Section 2.2. Section 3.6.2 describes the methodology used to obtain the results shown in section 2.2

16. Page 12, Line 384-385. Both RSDr and RSDR were done on 2 replicates, this need to be carefully checked whether is meaningful enough for method validation. 

Following the reviewer's advice, we have been decided to repeat some of these analyses to study precision by increasing the number of replicates. In this way, it increased to n=5 for RSDr and therefore to n=10 or 12 for RSDR. This number of replications is equal (for RSDr) to or even greater (for RSDR) than that employed, for example  by Villar-González et al. 2011 in a single- laboratory validation for the determination of lipophilic toxins by LC-MS/MS (Same answer as in question 8).

Reviewer 2 Report

Overall comment on “Development of a fast liquid chromatography coupled to mass spectrometry method (LC-MS/MS) to determine fourteen lipophilic shellfish toxins based on fused–core technology: In-house validation”.

The authors have developed a rapid LC-MS/MS method based on fused-core technology to determine 14 lipophilic shellfish toxins. The work is new and could be useful for rapid and multi-residue determination of lipophilic shellfish toxins. However, the English writing skill needs to be significantly improved, proper arrangement of sections and uniformity in writing needs to be developed. Therefore, I won’t suggest accepting the paper in its current format.

  • There are many English language writing issues (structural, grammatical and others).
  • Authors need to be attentive towards minute details in writing.
  • Lack of uniformity (ex: SPX1, 13desmSPXC, 13 desmethylspirolide C, 13-desmeSPXC).

Some section-wise comments are reflected below:

 Abstract:

  • “LC-MS/MS” is not the short form of liquid chromatography-mass spectrometry methodology. Please revise. Revise the same at Journal Title as well.
  • “Total single run”? It could be just mentioned as a single run.
  1. Introduction:
  • Please rephrase: “….that include okadaic acid (OA)…and gymnodimines (GYMs))”. Also. note the use of double parenthesis.
  • Change “..any part edible” to “any edible part”.
  • First-time use of abbreviations such as “LC-MS/MS” in the text must include its full form.
  • Check the format of citation: “..Gerssen, Mulder, McElhinney and de Boer [19]” and ..”
  • “..a 9-min chromatogram..”, is it correct? It should be 9 min chromatographic run.
  • “The characteristics are especially….batches”: Too long sentence, please consider making it more clear.
  • Why there is no full form of “OA, DTX1, DTX2,…..GYMA, 13,19didesmSPXC and 20Methyl 75 SPXG” at the very beginning?
  1. Results and Discussion
  • Many structural and writing issues (for example “..LC-MS/MS method developed..”, “..min 2.3..”, “..[17, 21-23, 25, 26]”, “Figure 1. ..obtained from with..”) and many more.
  • Figure 1: What is “level 6 of the calibration curves”?

2.1. LOQ/LOD

  • “..high sensitivity” or “highly sensitive”?
  • “(Qtrap 6500+)”, please provide full details of the company and all.
  • What is “priori”?

2.2. Selectivity/Specificity

  • “An important matrix…reported by [26]”, check citation format, it can’t be completed by “by and the citation number”.
  • “An intense signal extract was…in Moreiras et al. [22]”, “in” or “by” Moreiras?

2.3. Linearity

  • Check the use of r2 and r2.
  • Why the linearity for YTX can’t be achieved as of r2 = 0.99? Can you justify or repeat the experiment to eliminate the error?

2.4. Precision: Repeatability and Reproducibility

  • “Except level 1 for… (15–20%) (Table 5)”, will advise simplifying long sentences that will be easy to follow (breakdown into two).

2.5. Accuracy and recovery

  • “…with McCarron et al. 2017 [34]”, check citation style.
  1. Materials and Methods
  • This section is ideally placed before results and discussion.
  • What is “higher grade”? Be specific.

3.4. Samples extraction and hydrolysis

  • “Raw samples: For..”, is not the ideal writing pattern.

3.5. LC–MS/MS procedure

  • “…are shown in 326 Table 1..”, Where is Table 1 and also 2? I can’t see it in the paper. It’s better to mention whether it was SRM, MRM and to mention parent, daughter, and the qualifier ions.
  • The paper starts with “Table 3”, rather than “Table 1”.

Author Response

Reviewer 2

The authors have developed a rapid LC-MS/MS method based on fused-core technology to determine 14 lipophilic shellfish toxins. The work is new and could be useful for rapid and multi-residue determination of lipophilic shellfish toxins. However, the English writing skill needs to be significantly improved, proper arrangement of sections and uniformity in writing needs to be developed. Therefore, I won’t suggest accepting the paper in its current format.

  • There are many English language writing issues (structural, grammatical and others).
  • Authors need to be attentive towards minute details in writing.
  • Lack of uniformity (ex: SPX1, 13desmSPXC, 13 desmethylspirolide C, 13-desmeSPXC).

English language, drafting and uniformity in terms have been improved

Some section-wise comments are reflected below:

 Abstract:

  • “LC-MS/MS” is not the short form of liquid chromatography-mass spectrometry methodology. Please revise. Revise the same at Journal Title as well.

We prefer to maintain LC-MS/MS because is more specific than simply LC-MS, describes the method better, and is widely used in scientific literature. In fact, a search of “LC-MS/MS” in Google Scholar gives 1,280,000 references

  • “Total single run”? It could be just mentioned as a single run.

“Total” term has been removed

  1. Introduction:
  • Please rephrase: “….that include okadaic acid (OA)…and gymnodimines (GYMs))”. note the use of double parenthesis.

Sentence has been rewritten:

“Lipophilic toxins are natural metabolites produced by dinoflagellates which can be extracted from bivalve tissues using organic solvents. Structurally, they belong to five different groups (Figure 1): okadaic acid, including okadaic acid (OA), and dinophysistoxins (DTXs); azaspiracids (AZAs); pectenotoxins (PTXs); yessotoxins (YTXs); and cyclic imines (CIs), which include spirolides (SPXs), pinnatoxins (PnTXs), pteriatoxins (PtTXs), and gymnodimines (GYMs).”

  • Change “..any part edible” to “any edible part”.

Corrected

  • First-time use of abbreviations such as “LC-MS/MS” in the text must include its full form.

Corrected

  • Check the format of citation: “..Gerssen, Mulder, McElhinney and de Boer [19]” and ..”

Corrected here and throughout the manuscript

  • “..a 9-min chromatogram..”, is it correct? It should be 9 min chromatographic run.

Corrected

“The characteristics are especially….batches”: Too long sentence, please consider making it more clear.

Sentence has been rewritten as follows:

“These characteristics are especially interesting for monitoring systems because: a) working with lower pressures reduces the stress of the chromatographic equipment making it them less prone to failures; and b) using larger particles reduces the risk of aggregation of compounds in the column head which can interfere the analyses during long chromatographic batches.”

  • Why there is no full form of “OA, DTX1, DTX2,…..GYMA, 13,19didesmSPXC and 20Methyl 75 SPXG” at the very beginning?

Full forms of toxins have been included along the introduction section

  1. Results and Discussion
  • Many structural and writing issues (for example “..LC-MS/MS method developed..”, “..min 2.3..”, “..[17, 21-23, 25, 26]”, “Figure 1. ..obtained from with..”) and many more.

Wrong structural and writing issues corrected

  • Figure 1: What is “level 6 of the calibration curves”?

To know the concentrations of level 6 of calibration does not provide any information to the chromatogram, so we decided to eliminate it to avoid confusion.

2.1. LOQ/LOD

  • “..high sensitivity” or “highly sensitive”?

“high sensitivity” replaced by “highly sensitive”

  • “(Qtrap 6500+)”, please provide full details of the company and all.

Information about company has been provided

  • What is “priori”?

“a priori” removed

2.2. Selectivity/Specificity

  • “An important matrix…reported by [26]”, check citation format, it can’t be completed by “by and the citation number”.

Sentence has been removed

  • “An intense signal extract was…in Moreiras et al. [22]”, “in” or “by” Moreiras?

Sentence has been removed

2.3. Linearity

  • Check the use of r2 and r2.

Checked and corrected when necessary

  • Why the linearity for YTX can’t be achieved as of r= 0.99? Can you justify or repeat the experiment to eliminate the error?

YTX determination usually has more error than other toxins probably because of its ions are present in single and double charged ions. The double charged ones are used for quantification, so, small changes in the proportion between the two ions can increase the error of the YTX determination.

2.4. Precision: Repeatability and Reproducibility

  • “Except level 1 for… (15–20%) (Table 5)”, will advise simplifying long sentences that will be easy to follow (breakdown into two).

Sentence has been rewritten as follows:

“The obtained RSDr and RSDR values ranged from 3.1 to 21.9 % and 3.0 to 17.4% respectively (Table 4) (except for level 1 of GYMA and 13desmSPXC). These values comply with IUPAC, FDA and SANCO guidelines (15–20%) confirming the good precision of the method.”

2.5. Accuracy and recovery

  • “…with McCarron et al. 2017 [34]”, check citation style.

Reference has been eliminated.

  1. Materials and Methods
  • This section is ideally placed before results and discussion.

Following the editor's instruction, we have to keep it in its current position

What is “higher grade”? Be specific.

The first sentence has been removed.

3.4. Samples extraction and hydrolysis

  • “Raw samples: For..”, is not the ideal writing pattern.

Rewritten whole paragraph:

“Approximately 100 g of soft tissues of each tested bivalve (mussels, clams and oysters), not containing toxins (blank), were homogenized. For OA, DTXs, PTX2 and AZAs, aliquots of these blank samples were spiked with certified reference solutions to concentrations equivalent to LOQ, 0.5, 1, and 1.5 times the regulated levels (Table 10). For YTX and HomoYTX the concentrations were lower because their very high regulation limit makes it impossible to attain the required concentration without applying some concentration step to the currently available reference material (Table 10). Emerging toxins (13desmSPXC, GYMA, 13,19didesmSPXC, 20MethylSPXG and PnTXG) do not have regulated limits in the EU legislation so, fortification levels have been arbitrary established (Table 10). The freeze-dried reference material CRM-FDMT has been reconstituted following the procedure recommended by the manufacturer (NRC-CNRC).”

3.5. LC–MS/MS procedure

  • “…are shown in 326 Table 1..”, Where is Table 1 and also 2? I can’t see it in the paper. It’s better to mention whether it was SRM, MRM and to mention parent, daughter, and the qualifier ions.

Tables have been correctly numbered, and the quantifier and qualifier transitions are indicating in table 8.

  • The paper starts with “Table 3”, rather than “Table 1”.

Corrected

Round 2

Reviewer 2 Report

The paper can be accepted in its current form. The authors have addressed and revised the manuscript based on suggestions.